# Research on Motion Control and Wafer-Centering Algorithm of Wafer-Handling Robot in Semiconductor Manufacturing

**DOI:** 10.3390/s23208502

**Published:** 2023-10-16

**Authors:** Bing-Yuan Han, Bin Zhao, Ruo-Huai Sun

**Affiliations:** 1College of Information and Electrical Engineering, China Agricultural University, Beijing 110819, China; 2021312100206@cau.edu.cn; 2SIASUN Robot & Automation Co., Ltd., Shenyang 110168, China; 3School of Information Science & Engineering, Northeastern University, Shenyang 110819, China

**Keywords:** Integrated Circuit (IC), handling robot, motion control, Active Wafer Centering algorithm

## Abstract

This paper studies the AWC (Active Wafer Centering) algorithm for the movement control and wafer calibration of the handling robot in semiconductor manufacturing to prevent wafer surface contact and contamination during the transfer process. The mechanical and software architecture of the wafer-handling robot is analyzed first, which is followed by a description of the experimental platform for semiconductor manufacturing methods. Secondly, the article utilizes the geometric method to analyze the kinematics of the semiconductor robot, and it decouples the motion control of the robot body from the polar coordinates and joint space. The wafer center position is calibrated using the generalized least-square inverse method for AWC correction. The AWC algorithm is divided into calibration, deviation correction, and retraction detection. These are determined by analyzing the robot’s wafer calibration process. In conclusion, the semiconductor robot’s motion control and AWC algorithm are verified through experiments for correctness, feasibility, and effectiveness. After the wafer correction, the precision of AWC is <± 0.15 mm, which meets the requirements for transferring robot wafers.

## 1. Introduction

In the semiconductor equipment manufacturing industry, which has an exceptional working environment, the trajectory planning performance, reliability, and control accuracy of the wafer-handling robot are subject to strict requirements [1,2,3,4,5]. The semiconductor manufacturing process is exact and must be conducted in a strictly controlled environment. These environmental requirements must satisfy multiple core parameters like temperature, pressure, humidity, and cleanliness to ensure that the produced devices meet the anticipated performance and quality requirements. As the global semiconductor industry continues to expand, the market for semiconductor robots has expanded domestically and internationally [6]. Developing a semiconductor handling robot with China’s independent intellectual property rights and critical technologies is crucial to fulfilling the needs of the domestic semiconductor equipment manufacturing market [7,8]. The semiconductor industry must adhere to strict environmental standards to guarantee stability and consistency in the manufacturing process. In recent years, research on motion control and semiconductor wafer transfer methods has made significant progress, which has been demonstrated in many kinds of literature. Reference [9] reduced the wafer delay time at each step as much as possible. Xion, Pan, and Qiao propose three algorithms with polynomial complexity to assign the robot idle time as robot waiting time so that the wafer delay time in PMs can be reduced as much as possible. This work has a practical value. Reference [10] proposes a novel network WaferSegClassNet (WSCN) based on encoder–decoder architecture, which can solve the size of defects in semiconductor wafers. In the semiconductor wafer manufacturing process, the wafer goes through a series of continuous steps. Reference [11] presents a systematic approach to distributing wafers to maximize wafer yield while meeting predetermined target productivity levels. Reference [12] describes a wafer alignment algorithm that reduces the number of sensors to obtain the relative distance between the wafer center and the manipulator. It can be designed with only three sensor data, which is less than the four sensors of traditional algorithms. Therefore, the alignment algorithm proposed in this paper has the advantages of low design cost and low computing power.

More related research must be conducted on Active Wafer Centering abroad, and the AWC algorithm requires only two sensors to obtain the relative distance between the wafer center and the manipulator. This paper focuses on the motion control and AWC wafer calibration algorithm for semiconductor equipment handling robots, aiming to achieve fast, stable, and efficient transmission and operation in the manufacturing environment [13,14,15,16,17,18,19]. The paper analyzes the transmission mechanism and kinematics of the semiconductor handling robot using the geometric method, and the results form the basis of the AWC algorithm. The wafer for AWC (Active Wafer Centering) correction calibration is calculated using the generalized least squares inverse method based on the position of the finger center triggering the sensor twice. The wafer AWC algorithm consists of three processes: calibration, deviation correction, and retraction detection. This algorithm aims to complete wafer alignment quickly, improving the efficiency and quality of semiconductor production. Wafer alignment can be performed quickly, substantially increasing semiconductor production efficiency and quality. Wafer AWC technology is widely used in semiconductor production lines and has become essential for improving production efficiency and quality. Lastly, an experimental system is established, and the proposed algorithm is validated through experiments. 

## 2. Overview of Robot Systems

### 2.1. Mechanical Parameters

The robot for handling semiconductors has three active joints: one *Z*-axis to lift and lower independently, and two control axes, the rotating T-axis, and the extension R-axis. Figure 1 below shows that the two active joints controlled the blue (L) linkage rod and the red (R) linkage rod, respectively. The A/B hand is indicated in the figure. The position status of the robot in the figure should be set at zero and clockwise as positive (cf. Brooks). 

The rotation and extension motion of the semiconductor wafer-handling robot is realized by the combined motion of rod L and rod R, which has the following characteristics: The rotational motion of the robot is synthesized by the rotational motion of the connecting rods L and R at the same angular velocity, and the included angle between the connecting rod L and the connecting rod R remains unchanged at this time;The telescopic motion of the robot in a fixed position is composed of the rotational motion of the links L and R in opposite directions and at the same angular velocity. At this point, the arithmetic mean of the azimuth angles of the L and R links remains unchanged.When the angular rates of the L and R robot links are different, whether the direction of rotation is the same or not, macroscopically, it is reflected that the manipulator rotates while stretching.

### 2.2. Software Architecture

Figure 2 shows the composition of the software architecture system for the semiconductor robot and AWC flowchart. The AWC algorithm consists of three processes: calibration, deviation correction, and retraction detection. The software system uses a split-core control approach. The operating system for the semiconductor robot Robot Controller4.0 software supports MontaVista5.0 and is divided into a kernel layer (libraries), a platform layer, and an application layer. MontaVista5.0 Linux is the currently available embedded Linux development solution that supports the broadest range of CPU architectures, motherboards, and system-level platforms. It also provides a complete set of tools and deployable runtime components. The system features multiple single-core processors, each running one of the motion control algorithms, robotics algorithms, robot tasking, and robot referencing. In addition, there is data interaction with shared memory for robot variables and communication methods.

The kernel layer (library) is the RC (robot controller) general algorithm library, including FERAM (ferroelectric library), INST (instruction library), MODEL (model library), PLAN (planning library) and AWC (deviation correction algorithm).The platform layer is RC general software architecture, which is related to the RC system composition, including the FRAME library, BASE library, COMMU library, and TEACHBOX library.The application layer is the RC robot application system, including an atmospheric robot ATM, large load machine MHD, FPD, direct drive robot MAG and so on.Direct-drive robot application, mainly clean direct-drive robot application functions, including an application interface display, communication with the host computer, application instructions, application model, power-up logic of the application and other functions.

## 3. Motion Control of Handling Robot

Semiconductor handling robots typically have fewer degrees of freedom, with the main motions being rising (Z), stretching (R), and rotating (T). Due to the compact design of their mechanisms, they typically have kinematic coupling between multiple joints, which makes it necessary to implement decoupled control. 

### 3.1. Establishment of Polar Coordinate System for Handling Robot

Since the *Z*-axis of the robot is controlled independently as a dimension and does not need to involved in the kinematic calculation, it is possible to establish a planar coordinate system for calculations by rotating and telescoping the motion axes only. As shown in Figure 1, the polar coordinate system OX and the right-angle coordinate system $X_{0}OY_{0}$ are both fixed to the earth, defining the clockwise turning angle of the link L relative to the zero position as $\theta_{1}$ and the clockwise turning angle of the link R relative to the zero position as $\theta_{2}$. $\theta_{1}$, $\theta_{2}$: that is, respectively, the control motor rotation angle of the two connecting rods L and R in an RC system. 

### 3.2. Geometric Forward Kinematics (Joint Coordinates to Polar Coordinates)

Figure 3 shows the geometric schematic of the telescopic motion mechanism of a semiconductor handling robot’s A/B hand. Here, OE and OF refer to the connecting rods L and R, respectively. Since semiconductor handling robots have fewer degrees of freedom, operations using geometric methods usually have less computational effort, and results can be obtained quickly by utilizing geometric features in the robot mechanism. 

A hand extension: The figure above shows the state of the A hand extension of the semiconductor handling robot. OE and OF represent the mid-plumb line of AB and CD, respectively. The points of intersection of OP with AD and BC are denoted by J and K, respectively. The mid-plumb line of AD is denoted by OJ, while OK represents the mid-plumb line of BC. Drawing a plumb line from points, B and C intersect with EF at points U and V. Based on this, the following geometric relationship exists: $$\left\{\begin{array}{c} \bar{BM}=\bar{NC}=d_{1}\\ \bar{OE}=\bar{OF}=d_{2}\\ \bar{MP}=\bar{PN}=\bar{GJ}=\bar{JH}=\bar{QK}=\bar{KR}=d_{3}\\ \bar{AE}=\bar{EB}=\bar{CF}=\bar{FD}=d_{4}\\ \bar{BU}=\bar{CV}=\bar{JK}=\bar{GQ}=\bar{HR}\\ BU⊥EF,CV⊥EF \end{array}\right.$$

When the connecting rod L is rotated clockwise (positive angle) in relation to the connecting rod R, the semiconductor handling robot exhibits a state in which the A hand is extended. At this point, $∠EOF=180^{\circ }-(\theta_{1}-\theta_{2})$. 

Defining $\gamma =(\theta_{1}-\theta_{2})/2$, then $∠EOF=2(90^{\circ }-\gamma$). Here, $\gamma \in \left[0^{\circ },90^{\circ }\right)$, as shown in the figure above: (1)$$\left\{\begin{array}{c} ∠EOP=∠POF=∠BEU=∠CFV=90^{\circ }-\gamma \\ \bar{EJ}=\bar{OE}\times sin\left(∠EOP\right)=d_{2}\cos(\gamma )\\ \bar{EU}=\bar{EB}\times cos\left(∠BEU\right)=d_{4}\sin(\gamma ) \end{array}\right.$$

Since $\bar{BQ}=\bar{UG}=\bar{EJ}-\bar{GJ}-\bar{EU}=d_{2}cos\left(\gamma \right)-d_{4}sin(\gamma )-d_{3}$. Thus,
(2)$$∠QMB=arcsin\left(\bar{BQ}/\bar{BM}\right)$$

The length of the OP represents the coordinate of the R-axis, that is: (3)$$\left\{\begin{array}{c} \bar{OP}=\bar{OJ}+\bar{JK}+\bar{KP}\\ =\bar{OE}\times \cos\left(∠EOP\right)+\bar{EB}\times \sin\left(∠BEU\right)+\bar{BM}\times \cos\left(∠QMB\right)\\ =d_{2}\sin\left(\gamma \right)+d_{4}\cos\left(\gamma \right)+d_{1}\cos\left(f\right) \end{array}\right.$$

Among them,
(4)$$f=arcsin\left((d_{2}cos(\gamma )-d_{4}sin(\gamma )-d_{3})/d_{1}\right)$$

B hand extension: The figure above shows the status of the B hand extension of the semiconductor handling robot. OE denotes the mid-plumb line of AB, and OF denotes the mid-plumb line of CD. J′ and K′ are the points where OP′ intersects AD and BC, respectively. OJ′ represents the midpoint of the perpendicular from A to D, while OK′ represents the midpoint of the perpendicular from B to C. The vertical line drawn from points B and C intersects EF at points U′ and V′, respectively. Based on this, the following geometric relationship exists: $$\left\{\begin{array}{c} \bar{AM\prime }=\bar{N\prime D}=d_{1}\\ \bar{OE}=\bar{OF}=d_{2}\\ \bar{M\prime P\prime }=\bar{P\prime N\prime }=\bar{G\prime J}\prime =\bar{J\prime H\prime }=\bar{Q\prime K\prime }=\bar{K\prime R\prime }=d_{3}\\ \bar{AE}=\bar{EB}=\bar{CF}=\bar{FD}=d_{4}\\ \bar{AU\prime }=\bar{DV\prime }=\bar{J\prime K\prime }=\bar{G\prime Q\prime }=\bar{H\prime R\prime }\\ AU\prime ⊥EF,CV⊥EF \end{array}\right.$$

When the connecting rod L rotates counterclockwise (at a negative angle) relative to the connecting rod R, the semiconductor handling robot exhibits a state in which the B hand is extended. At this point, we define $\gamma =(\theta_{1}-\theta_{2})/2$, then $∠EOF=2(90^{\circ }+\gamma )$. Here, $\gamma \in \left(-90^{\circ },0^{\circ }\right]$, which corresponds to the figure above: (5)$$∠EOP\prime =∠P\prime OF=∠AEU\prime =∠DFV\prime =90^{\circ }+\gamma$$
(6)$$\left\{\begin{array}{c} \bar{EK\prime }=\bar{OE}\times \sin\left(∠EOP\prime \right)=d_{2}\cos(\gamma )\\ \bar{EU\prime }=\bar{AE}\times \cos\left(∠AEU\prime \right)=-d_{4}\sin(\gamma ) \end{array}\right.$$

Among them, $\bar{OP\prime }=\bar{OK\prime }+\bar{K\prime J\prime }+\bar{J\prime P\prime }$, $\bar{AG\prime }=\bar{U\prime Q\prime }=\bar{EK\prime }-\bar{Q\prime K\prime }-\bar{EU\prime }$. Hence, we obtain the equation:(7)$$∠G\prime M\prime A=arcsin\left(\bar{AG\prime }/AM\prime \right)$$

The length of $\bar{OP\prime }$ determines the coordinate of the R-axis and can be calculated as follows: (8)$$\left\{\begin{array}{c} \bar{OP\prime }=\bar{OK\prime }+\bar{K\prime J\prime }+\bar{J\prime P\prime }\\ =\bar{OE}\times \cos\left(∠EOP\prime \right)+\bar{AE}\times \sin\left(∠AEU\prime \right)+\bar{AM\prime }\times \cos\left(∠G\prime MA\right)\\ =-d_{2}\sin\left(\gamma \right)+d_{4}\cos\left(\gamma \right)+d_{1}\cos\left(g\right) \end{array}\right.$$

Among them,
(9)$$g=arcsin\left((d_{2}cos(\gamma )+d_{4}sin(\gamma )-d_{3})/d_{1}\right)$$

T-axis rotation: At the zero position, the A axis of the semiconductor handling robot is oriented in the positive direction of the polar axis OX at an azimuth angle of $0^{\circ }$ and in the positive direction of the $OY_{0}$ axis in the rectangular auxiliary coordinate system at an azimuth angle of $90^{\circ }$. At this point, the azimuth of the L link (OE) is $-90^{\circ }+\theta_{1}$ in the polar coordinate system, and it is $\theta_{1}$ in the auxiliary rectangular coordinate system whereas the azimuth of the R-link (OF) is $90^{\circ }+\theta_{2}$ in the polar coordinate system, and it is $180^{\circ }+\theta_{2}$ in the auxiliary rectangular coordinate system. The angle bisector of ∠EOF indicates the direction of hand A of the manipulator. In the polar coordinate system, the azimuth of the manipulator is $(\theta_{1}+\theta_{2})/2$. In the auxiliary rectangular coordinate system, it is $90^{\circ }+(\theta_{1}+\theta_{2})/2$. Similarly, the same can be obtained for the B manipulator hand. 

Thus, the plane polar coordinates value for the manipulator A/B hand is: (10)$$\left\{\begin{array}{c} \theta_{T}=(\theta_{1}+\theta_{2})/2\\ r=\arcsin\left((d_{2}cos(\gamma )-d_{4}sin(\gamma )-d_{3})/d_{1}\right)\\ R=side\times d_{2}\sin\left(\gamma \right)+d_{4}\cos\left(\gamma \right)+d_{1}\cos\left(r\right) \end{array}\right.$$

The solution to the inverse problem for the robot that handles semiconductors is simpler and will not be restated in this context. 

## 4. Automatic Wafer Alignment

Wafer AWC, known as Active Wafer Centering, is an advanced semiconductor wafer compensation technology that utilizes optical sensors and an automated control system to position automatically and center-align wafers. Wafer AWC technology has higher precision, faster speed, and more stable performance than the traditional mechanical manual alignment method. Specifically, the principle of wafer AWC technology is to scan and measure the surface of the wafer through optical sensors to obtain real-time information on the position, rotation angle, and deviation of the wafer and transmit this information to the automatic control system for analysis and calculation. The automatic control system will accurately control the robotic arm’s movement according to the wafer’s real-time state so that the wafer goes back to the correct position and angle to ensure chip preparation and processing accuracy. The AWC algorithm consists of three parts: 

Calibration: Position the robot finger center to match the wafer center. Controlling the robot to perform the high telescoping operation of the work station captures the spatial position of the manipulator when the sensor is triggered. From this, the positions of the two sensors in the robot’s base coordinate system $\left(P_{xS},P_{yS}\right)$, $\left(P_{xR},P_{yR}\right)$ are calculated. 

Correcting deviation: Determine the deviation between the wafer center and the robot finger center. By analyzing the position of the dual sensors and considering the deviation that occurs during wafer placement, we can determine the deviation of the wafer relative to the robot center $\left(D_{x},D_{y}\right)$ in the robot coordinate system. 

Detecting retraction: During the return trip, the wafer retrieved from the station will activate the sensor twice and record the position of these two points. By utilizing the spatial position information of the sensor, we can measure the required $\left(\Delta R,\Delta T\right)$ to correct the wafer as the deviation value. 

### 4.1. Calibrated NOTCH Point Filtering

The approach for NOTCH discrimination is as follows: Notably, in Figure 4, the opening of the NOTCH is less than 2.73 mm. Based on the theoretical data, a certain margin is considered, and the NOTCH threshold is set to 4 mm (NOTCH_THRES = 4 mm). In practical applications, the two strings have sizes of 0.8 mm and 1.2 mm, respectively.

The criterion is as follows: for any three consecutive trigger points a, b, and c, the sufficient condition for passing through the NOTCH is D_ab_ < NOTCH_THRES, and D_bc_ < NOTCH_THRES. 

Thus, the general approach follows:

The wafer on the manipulator passes through the sensor to ensure the presence of a trigger point.

The number of trigger points is discussed in various cases: Four. If NOTCH is present among the four, report an error. Otherwise, calculate the AWC deviation normally.Five. If NOTCH is present among the five, report an error. After that, determine if the 4th and 5th trigger points may be located at NOTCH, and report an error if they may be located there. If NOTCH is absent or the trigger points are not located at NOTCH, discard the 5th point and use the first 4 points for AWC deviation calculation.Six and above. For values 6 and above, exclude all departure points except for the first six. Then, check if each set of three consecutive points is activated by crossing NOTCH. If so, eliminate the data from those three points and use the remaining three points for calculating AWC deviation.Three and below. If the value is 3 or lower, there are not enough collection points for AWC calculation, and an error will be reported.

### 4.2. Sensor Calibration

The paper uses the KEYENCE Probe sensor of AWC, which has FINE, TURBO and HPS mode. Select the appropriate mode according to the product manual. The schematic diagram depicted in Figure 5 displays the trigger position and sensor position during AWC calibration. R and S are the sensors, and the circle is the wafer. The calibration procedure consists of the following steps: Select the workstation and place the wafer in the center of the manipulator, making sure that the center of the wafer and the center of the manipulator coincide.Control the robot and move the edge of the wafer to trigger the sensors four times in this sequence: the left sensor retracts twice and the right sensor retracts twice as the robot moves to the edge of the workstation.

In the Figure 5, the positions of the robot centers $A_{0}$ and $B_{0}$ should be recorded as $\left(\rho_{1},\theta_{1}\right)$ and $\left(\rho_{2},\theta_{2}\right)$, respectively, while the planar positions of the points $A_{0}$ and $B_{0}$ should be recorded as $\left(\rho_{1}s_{1},\rho_{1}c_{1}\right)$ and $\left(\rho_{2}s_{2},\rho_{2}c_{2}\right)$, where $s_{1},c_{1},s_{2},andc_{2}$ are the sine and cosine of $\theta_{1}$ and $\theta_{2}$. Record the positions of robot centers $C_{0}$ and $D_{0}$ as $\left(\rho_{3},\theta_{3}\right)$ and $\left(\rho_{4},\theta_{4}\right)$, respectively, and their planar positions as $\left(\rho_{3}s_{3},\rho_{3}c_{3}\right)$ and $\left(\rho_{4}s_{4},\rho_{4}c_{4}\right)$, where $s_{3}$, $c_{3}$, $s_{4}$, and $c_{4}$ are the sine and cosine of the angles $\theta_{3}$ and $\theta_{4}$. The positions of the sensors are labeled as points S and R. The vertical diameter theorem in plane geometry states that point N is the intersection of perpendicular segments $A_{0}B_{0}$ and SS′. Likewise, line segments $C_{0}D_{0}$ and RR′ intersect perpendicularly at point M, according to the same theorem. 

**Figure 5 sensors-23-08502-f005:**
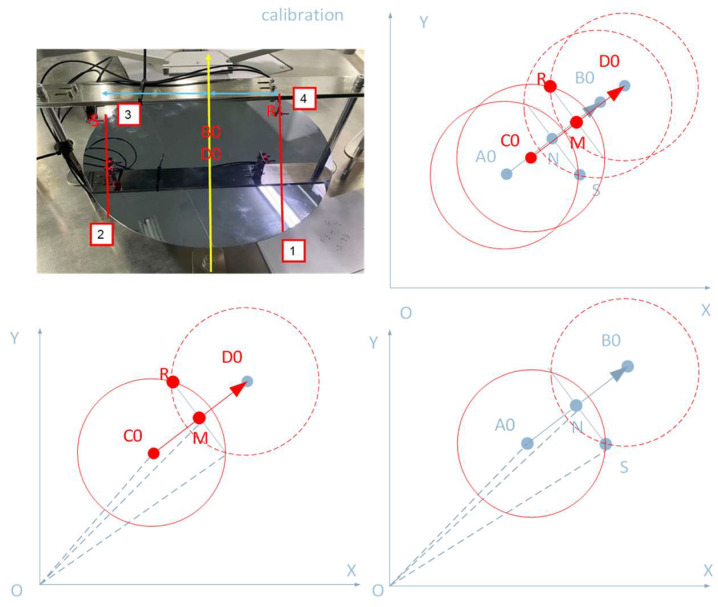
Two-dimensional (2D) geometry of trigger and sensor positions during calibration of AWC calibration.

The sensor on the right is calculated using the following: (11)$$\left\{\begin{array}{c} \vec{e_{A_{0}B_{0}}}=\frac{\vec{A_{0}B_{0}}}{\left|A_{0}B_{0}\right|}=\left(\begin{array}{c} e_{x}\\ e_{y} \end{array}\right)\\ \vec{e_{NS}}=\left(\begin{array}{c} e_{y}\\ -e_{x} \end{array}\right),\vec{e_{NS\prime }}=\left(\begin{array}{c} -e_{y}\\ e_{x} \end{array}\right) \end{array}\right.$$

Among them,
(12)$$\left|NS\right|=\left|NS\prime \right|=\sqrt{R_{wafer}^{2}-{\left(\frac{\left|A_{0}B_{0}\right|}{2}\right)}^{2}}$$

The coordinates of the sensor are as follows: (13)$$\left(\begin{array}{c} P_{xS}\\ P_{yS} \end{array}\right)=\frac{1}{2}\left(\begin{array}{c} R_{1}s_{1}\\ R_{1}c_{1} \end{array}\right)+\frac{1}{2}\left(\begin{array}{c} R_{2}s_{2}\\ R_{2}c_{2} \end{array}\right)+side\times \left|d_{S}\right|\left(\begin{array}{c} e_{y}\\ -e_{x} \end{array}\right)$$

To obtain the coordinates of the left sensor position, the same process used for the right sensor can be followed: (14)$$\left(\begin{array}{c} P_{xR}\\ P_{yR} \end{array}\right)=\frac{1}{2}\left(\begin{array}{c} R_{3}s_{3}\\ R_{3}c_{3} \end{array}\right)+\frac{1}{2}\left(\begin{array}{c} R_{4}s_{4}\\ R_{4}c_{4} \end{array}\right)+side\times \left|d_{R}\right|\left(\begin{array}{c} e_{y}\\ -e_{x} \end{array}\right)$$

The sensor orientation parameter in the workstation is denoted by the variable ‘side’. When observing the workstation from the base of the manipulator, if the sensor is located at point S on the right side of the observing direction, then the ‘side’ parameter is set to 1. Vice versa, it is set to −1. 

The dual sensor positions follow: (15)$$\left\{\begin{array}{c} \left(\begin{array}{c} P_{xS}\\ P_{yS} \end{array}\right)=\frac{1}{2}\left(\begin{array}{c} R_{1}s_{1}\\ R_{1}c_{1} \end{array}\right)+\frac{1}{2}\left(\begin{array}{c} R_{2}s_{2}\\ R_{2}c_{2} \end{array}\right)+\left|d_{S}\right|\left(\begin{array}{c} e_{y}\\ -e_{x} \end{array}\right)\\ \left(\begin{array}{c} P_{xR}\\ P_{yR} \end{array}\right)=\frac{1}{2}\left(\begin{array}{c} R_{3}s_{3}\\ R_{3}c_{3} \end{array}\right)+\frac{1}{2}\left(\begin{array}{c} R_{4}s_{4}\\ R_{4}c_{4} \end{array}\right)+\left|d_{S}\right|\left(\begin{array}{c} e_{y}\\ -e_{x} \end{array}\right) \end{array}\right.$$

### 4.3. Generalized Inverse Method Correction for Least Squares 

#### 4.3.1. Calculation of Wafer Deviation in Finger Coordinate System

The deviation of the wafer in the finger coordinate system $\left(D_{x},D_{y}\right)$ should be calculated. During the wafer-picking process, the robot moves from the home state toward the workstation. The right sensor will sequentially trigger twice, corresponding to the finger centers of *A*_1_ and *B*_1_ and the wafer centers of *O*_1_ and *O*_2_, respectively. The left sensor, while not shown in the figure, will also sequentially trigger twice, corresponding to the finger centers of *C*_1_ and *D*_1_ and the wafer centers of *O*_3_ and *O*_4_, respectively. However, the left sensor is approximated with the right. 

Figure 6 shows the base coordinate system of the robot, which is named *XOY*. The finger coordinate system, $x_{1}$o$y_{1}$, in which *A*_1_ is located, is derived by rotating *XOY* clockwise by $\theta_{1}$ and then translating it by $\rho_{1}$ along the positive *Y*-axis. The coordinates of *O*_1_ in $x_{1}$o$y_{1}$ are $D_{x}$ and $D_{y}$. Therefore, the coordinates of *O*_1_ in *XOY* follow: (16)$$\left(\begin{array}{c} x_{O_{1}}\\ y_{O_{1}}\\ 1 \end{array}\right)=\left[\begin{array}{ccc} c_{1} & s_{1} & 0\\ -s_{1} & c_{1} & 0\\ 0 & 0 & 1 \end{array}\right]\left[\begin{array}{ccc} 1 & 0 & 0\\ 0 & 1 & \rho_{1}\\ 0 & 0 & 1 \end{array}\right]\left[\begin{array}{c} D_{x}\\ D_{y}\\ 1 \end{array}\right]$$

Through analogy, we can obtain the following: (17)$$\left\{\begin{array}{c} \left(\begin{array}{c} x_{1}\\ y_{1} \end{array}\right)=\left(\begin{array}{c} \rho_{1}s_{1}+D_{x}c_{1}+D_{y}s_{1}\\ \rho_{1}c_{1}-D_{x}s_{1}+D_{y}c_{1} \end{array}\right)\\ \left(\begin{array}{c} x_{2}\\ y_{2} \end{array}\right)=\left(\begin{array}{c} \rho_{2}s_{2}+D_{x}c_{2}+D_{y}s_{2}\\ \rho_{2}c_{2}-D_{x}s_{2}+D_{y}c_{2} \end{array}\right)\\ \left(\begin{array}{c} x_{3}\\ y_{3} \end{array}\right)=\left(\begin{array}{c} \rho_{1}s_{3}+D_{x}c_{3}+D_{y}s_{3}\\ \rho_{1}c_{3}-D_{x}s_{3}+D_{y}c_{3} \end{array}\right)\\ \left(\begin{array}{c} x_{4}\\ y_{4} \end{array}\right)=\left(\begin{array}{c} \rho_{2}s_{4}+D_{x}c_{4}+D_{y}s_{4}\\ \rho_{2}c_{4}-D_{x}s_{4}+D_{y}c_{4} \end{array}\right) \end{array}\right.$$

Since all four sensor coordinates lie on the same circle, it is possible to locate the center of the circle, which is the deviation of the wafer center from the finger center. It is possible to construct a system of equations based on the equal magnitudes of $\left|SO_{1}\right|$ and $\left|SO_{2}\right|$ as well as of $\left|RO_{3}\right|$ and $\left|RO_{4}\right|$: (18)$$F=\left\{\begin{array}{c} {\left(\rho_{1}s_{1}+D_{x}c_{1}+D_{y}s_{1}-x_{S}\right)}^{2}+\\ {\left(\rho_{1}c_{1}-D_{x}s_{1}+D_{y}c_{1}-y_{S}\right)}^{2}-R_{wafer}^{2}=0\\ {\left(\rho_{2}s_{2}+D_{x}c_{2}+D_{y}s_{2}-x_{S}\right)}^{2}+\\ {\left(\rho_{1}c_{2}-D_{x}s_{2}+D_{y}c_{2}-y_{S}\right)}^{2}-R_{wafer}^{2}=0\\ \begin{array}{c} {\left(\rho_{3}s_{3}+D_{x}c_{3}+D_{y}s_{3}-x_{R}\right)}^{2}+\\ {\left(\rho_{3}c_{3}-D_{x}s_{3}+D_{y}c_{3}-y_{R}\right)}^{2}-R_{wafer}^{2}=0\\ {\left(\rho_{4}s_{4}+D_{x}c_{4}+D_{y}s_{4}-x_{R}\right)}^{2}+\\ {\left(\rho_{4}c_{4}-D_{x}s_{4}+D_{y}c_{4}-y_{R}\right)}^{2}-R_{wafer}^{2}=0 \end{array} \end{array}\right.$$

We solve the equation mentioned above to determine the values of the unknown parameters $\left(D_{x},D_{y}\right)$.

**Figure 6 sensors-23-08502-f006:**
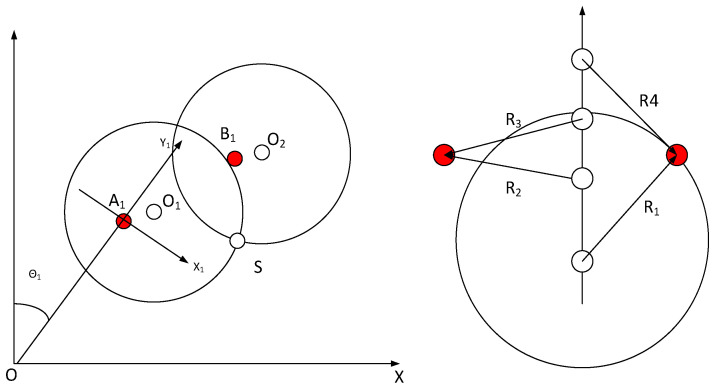
Schematic diagram of trigger position and sensor position during AWC correction.

#### 4.3.2. Generalized Inverse Method for Least Squares Solutions of Systems of Nonlinear Equations

In a real sensor network, the four sensor coordinate positions may not lie on the same circle due to factors like measurement errors and sensor errors. As a result, the ideal case system of Equation (6) becomes hyper-positive definite, and an exact solution cannot be obtained. To solve this issue, the system of hyper-positive definite equations can be approximated using the least squares method to allow localization calculations based on sensor positions in all cases. It is important to note that as this is a nonlinear system of equations, the generalized inverse matrix can be utilized to solve the optimization problem without constraints. This approach successfully solves the localization problem in sensor networks.
(19)$$\left\{\begin{array}{c} F^{\left(k\right)}={\left(f_{0}^{\left(k\right)},f_{1}^{\left(k\right)},\ldots ,f_{n-1}^{\left(k\right)}\right)}^{T}\\ f_{i}^{\left(k\right)}=f_{i}\left(D_{0}^{\left(k\right)},D_{1}^{\left(k\right)},\ldots ,D_{m-1}^{\left(k\right)}\right),i=0,1,\ldots ,n-1\\ f_{i}\left(D_{0},D_{1},\ldots ,D_{m-1}\right)=0,i=0,1,\ldots ,n-1,n⩾m \end{array}\right.$$

If m equals n, the system of nonlinear equations can be solved. Here, $\left(D_{0},D_{1}\right)$ equals $\left(D_{x},D_{y}\right)$ and m is equal to 2. The resultant Jacobi matrix is as follows: (20)$$A=\left[\begin{array}{cccc} \frac{\partial f_{0}}{\partial D_{0}} & \frac{\partial f_{0}}{\partial D_{1}} & \ldots & \frac{\partial f_{0}}{\partial D_{m-1}}\\ \frac{\partial f_{1}}{\partial D_{0}} & \frac{\partial f_{1}}{\partial D_{1}} & \ldots & \frac{\partial f_{1}}{\partial D_{m-1}}\\ \vdots & \vdots & & \vdots \\ \frac{\partial f_{m-1}}{\partial D_{0}} & \frac{\partial f_{m-1}}{\partial D_{1}} & \ldots & \frac{\partial f_{n-1}}{\partial D_{m-1}} \end{array}\right]$$

Here is the iterative formula to compute the least squares solution for the system of AWC corrective nonlinear equations: (21)$$D^{\left(k+1\right)}=D^{\left(k\right)}-\alpha_{k}Z^{\left(k\right)}$$

Here, $Z^{\left(k\right)}$ represents the solution obtained through the linear least squares method for the system of linear algebraic equations $A^{\left(k\right)}Z^{\left(k\right)}=F^{\left(k\right)}$. The value of $\alpha_{k}$ is the extremum of the unitary function $\sum_{i=0}^{m-1}\;{\left(f_{i}^{\left(k\right)}\right)}^{2}$, which is calculated using the method of rational extrema. This leads to a particular solution, revealing the deviation of the wafer center from the finger’s coordinate system.

### 4.4. Retraction Detection

To achieve alignment of the wafer center to the station, the T and R coordinates of the robot are deflected, resulting in ΔR for telescopic compensation and ΔT for rotational compensation. 

Right sensor compensation: As shown in the Figure 7 above, we can obtain the following:(22)$$OA_{1}=\sqrt{{\left|OO_{1}\right|}^{2}-{C_{1}O_{1}}^{2}}-{C_{1}A}_{1}$$

The coordinates of point *O*_1_ in the *XOY* plane follow: (23)$$\left\{\begin{array}{c} \left|OO_{1}\right|=RR_{1}\;\rho_{1}=OA_{1}\\ \left(\begin{array}{c} x_{O_{1}}\\ y_{O_{1}} \end{array}\right)=\left(\begin{array}{c} RR_{1}c_{2}\\ RR_{1}s_{2} \end{array}\right)=\left(\begin{array}{c} (\rho_{1}+D_{y})s_{1}+D_{x}c_{1}\\ (\rho_{1}+D_{y})c_{1}-D_{x}s_{1} \end{array}\right) \end{array}\right.$$

After derivation, we obtain $\theta_{2}=2tan(s2,c2)$, leading to the following: (24)$$\left\{\begin{array}{c} {\nabla R}_{r}=OA_{1}-RR_{1}\\ {\nabla T}_{r}=\theta_{2}-\theta_{1} \end{array}\right.$$

**Figure 7 sensors-23-08502-f007:**
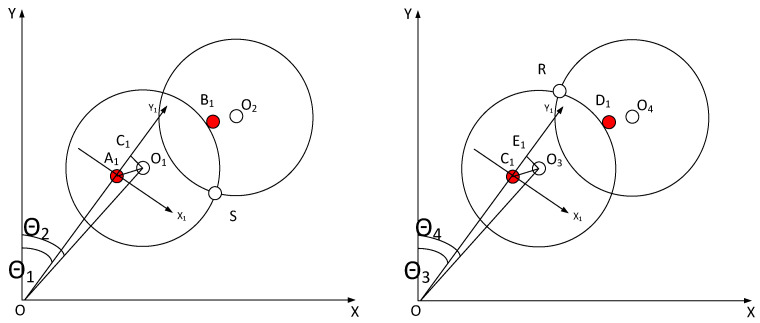
Schematic diagram of trigger position and sensor position during AWC retraction compensation.

Left sensor compensation: The retraction detection values ∇R and ∇T of the left sensor can be obtained using the same method as the right sensor.
(25)$$\left(\begin{array}{c} c_{4}\\ s_{4} \end{array}\right)=\left(\begin{array}{c} (\rho_{3}+D_{y})s_{3}/RR_{2}+D_{x}c_{3}/RR_{2}\\ (\rho_{3}+D_{y})c_{3}/RR_{2}-D_{x}s_{3}/RR_{2} \end{array}\right)$$

After derivation, we obtain $\theta_{4}=2tan(s4,c4)$, leading to the following: (26)$$\left\{\begin{array}{c} {\nabla R}_{l}=OC_{1}-RR_{2}\\ {\nabla T}_{l}=\theta_{4}-\theta_{3} \end{array}\right.$$

The values for expansion compensation, ΔR, and rotation compensation, ΔT, can be determined as follows: (27)$$\left\{\begin{array}{c} \Delta R=\left({\nabla R}_{r}+{\nabla R}_{l}\right)/2\\ \Delta T=\left({\nabla T}_{r}+{\nabla T}_{l}\right)/2 \end{array}\right.$$

After completing the three processes of calibration, correction, and retraction detection, AWC correction calibration is performed on the wafer. 

## 5. Experimental Results and Analysis

The validation platform for the semiconductor handling robot motion control and AWC algorithm is presented in Figure 8. The motion control and wafer calibration algorithms, proposed in this paper, operate on the Linux Ubuntu operating system. Thereafter, the validation of motion control and wafer calibration algorithms is discussed. 

### 5.1. Motion Control Experiment

The equations for the conic isometric helix are inputted into the offline programming software to verify both the positive and negative solutions for the kinematics as follows: (28)$$\left\{\begin{array}{c} x=(Rvt/H)\cos\omega t\\ y=(Rvt/H)\sin\omega t\\ z=vt \end{array}\right.$$

Here, H represents the height of the isometric helix of the conical plane; ω denotes the rotational angular velocity around the center axis; R signifies the radius of the isometric helix plane of the conical plane; and v denotes the ascending linear velocity. 

Figure 9 shows that the red curve represents the robot’s real trajectory $P=[\begin{array}{ccc} p_{x} & p_{y} & p_{z} \end{array}]$ taught using the helix equation, while the green color is the trajectory taught using inverse kinematics for finding joint values and positive kinematics for obtaining the reference trajectory value $P^{\prime }=[\begin{array}{ccc} p_{x}^{\prime } & p_{y}^{\prime } & p_{z}^{\prime } \end{array}]$. The plot of the collaborative robot’s end-demonstration trajectory P (red) agrees with the kinematic solution trajectory $P^{\prime }$ (green), thus demonstrating the correctness of the inverse kinematic modeling. 

We have experimentally verified the robot kinematic error and implemented the proposed inverse kinematic travel power consumption constraint algorithm in the robot controller. The experiment was successfully replicated during the demonstration. Figure 10 demonstrates the absolute kinematic error. The absolute error between the theoretical position value of the robot and the actual position value of the kinematics is approximately 0.0001 mm, which satisfies the requirements for engineering applications. 

### 5.2. Automatic Wafer Alignment Calibration Correction Experiment

To verify the accuracy of the automatic wafer alignment calibration algorithm, a handling robot was used to experiment with the algorithm in a real semiconductor industry environment. For random sampling analysis, the accuracies of two stations, namely STN2 (station two) and STN4 (replace station with STN after), have been measured. The position of STN2 is {900.75 mm, 46.47°} and that of STN4 is {931.26 mm, 1.21°}, respectively. Table 1 and Table 2 show the data collected in millimeters using the AWC algorithm for the calibration of the extended and retracted dual sensor positions. 

Table 3 shows the method of calculating wafer deviation (mm) in the finger coordinate system using the least squares-based generalized inverse. 

Table 4 displays the values for expansion compensation (ΔR in mm) and rotation compensation (ΔT in degrees). 

The robot verifies the wafers through three processes: calibration, deviation correction, and retraction detection. To perform the accuracy analysis, the accuracy of two stations, STN2 and STN4, is taken, and then the calculated data are collected. The correctness of the algorithm can be demonstrated by measuring the deviation of the wafer (Figure 11a,b).

### 5.3. Cyclic Automatic Wafer Alignment Error Experiment

The AWC algorithm has been analyzed and validated for accuracy data errors below a certain threshold. STN represents the position of the wafer box, and the black triangle represents the contact point of silicon wafer and the robot. The algorithm presented in this paper requires the use of three stations for this test, as depicted in Figure 12. 

The experimental test flow proceeds as follows. First, the STN2 wafer is picked up by hand B and transferred to STN3; then, hand A moves the STN3 wafer to STN4, and finally, hand B moves the STN4 wafer back to STN2. Figure 13, Figure 14 and Figure 15 illustrates the error data collected over 24 h of AWC calibration and correction. 

In practical applications where the working range exceeds 1000 mm, AWC achieves an accuracy of <±0.15 mm after correction. The primary source of error is the subdivision error of the code disk angle precision of the GPI, which may break down due to precision deviation because the etching process requires high accuracy to prevent errors. Poor positioning accuracy can hinder its use in the etching process industry. 

## 6. Conclusions

This paper discusses the issues of motion control and wafer alignment algorithms for robots used in semiconductor manufacturing. Regarding motion control, this paper proposes a geometric approach to analyze the kinematics of robots used for semiconductor handling. The suggested method decouples the body motion control into polar coordinates and joint space. Concerning the wafer calibration algorithm, this paper presents a silicon wafer calibration algorithm. The proposed method solves the nonlinear equation problem of wafer position correction through a generalized least squares inverse method called the AWC algorithm. The experiments demonstrate that the method proposed in this document achieves an absolute error of 0.0001 mm in kinematics and an accuracy of less than ±0.15 mm after AWC correction. These results confirm the effectiveness and validity of the proposed method. This study provides a basis for the practical implementation of semiconductor wafer-handling robots and serves as a reference for high-quality semiconductor manufacturing. 

## Figures and Tables

**Figure 1 sensors-23-08502-f001:**
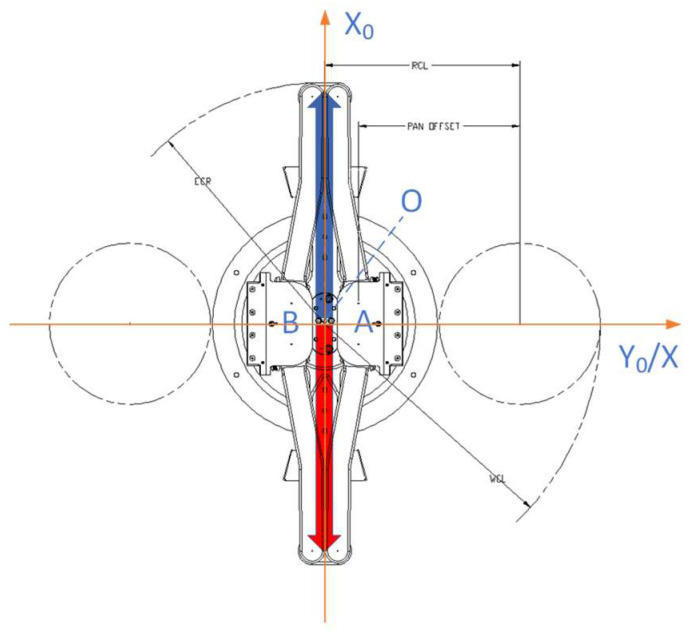
Schematic of active actuator and coordinate system of semiconductor handling robot.

**Figure 2 sensors-23-08502-f002:**
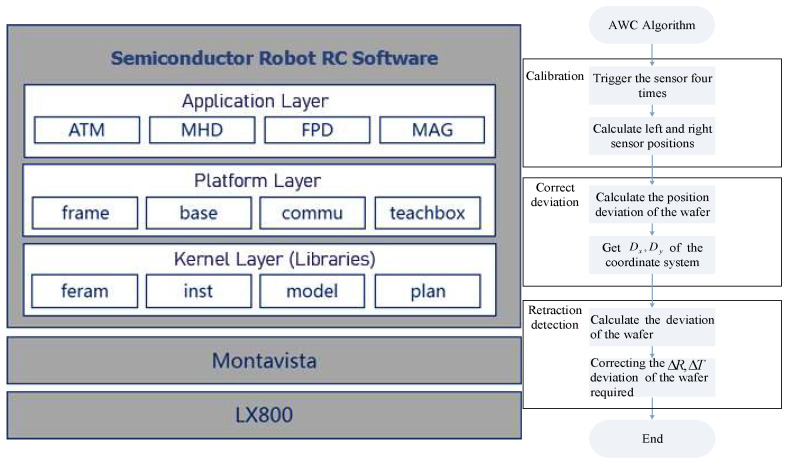
Software architecture of semiconductor robot and AWC flowchart.

**Figure 3 sensors-23-08502-f003:**
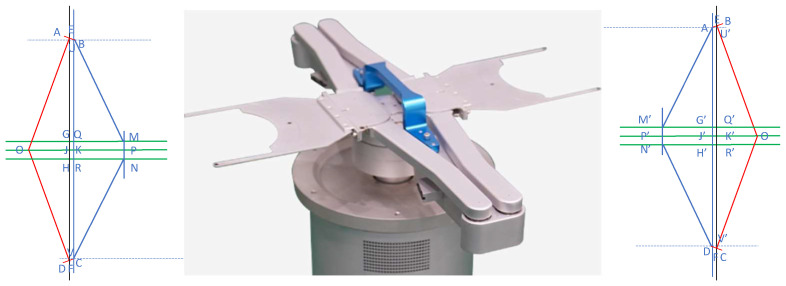
Geometric diagram of the changes in the A/B hand telescope mechanism.

**Figure 4 sensors-23-08502-f004:**
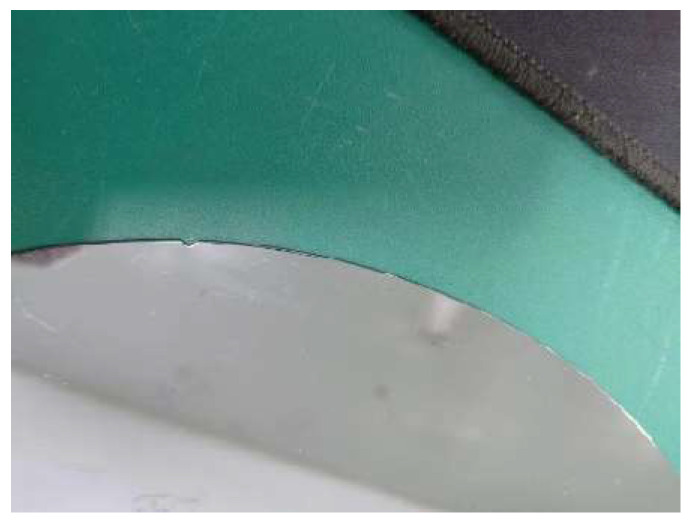
NOTCH discrimination.

**Figure 8 sensors-23-08502-f008:**
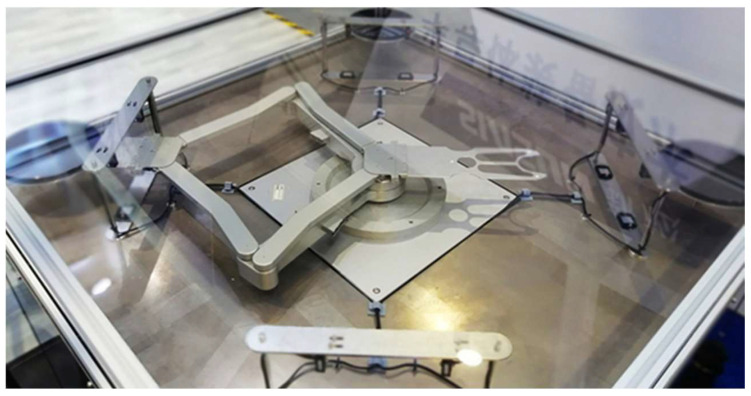
Algorithm validation platform for semiconductor-handling robots.

**Figure 9 sensors-23-08502-f009:**
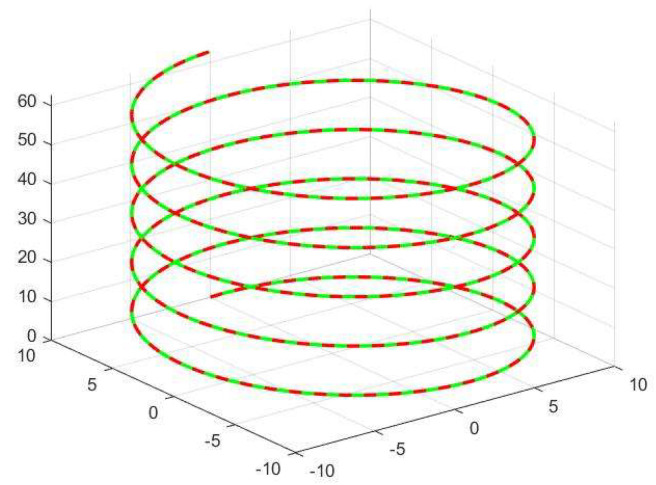
Kinematic validation.

**Figure 10 sensors-23-08502-f010:**
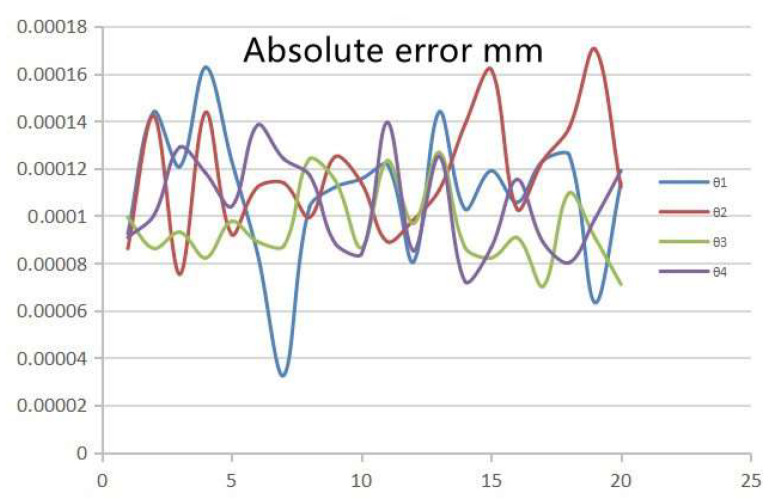
Schematic diagram of absolute kinematic error.

**Figure 11 sensors-23-08502-f011:**
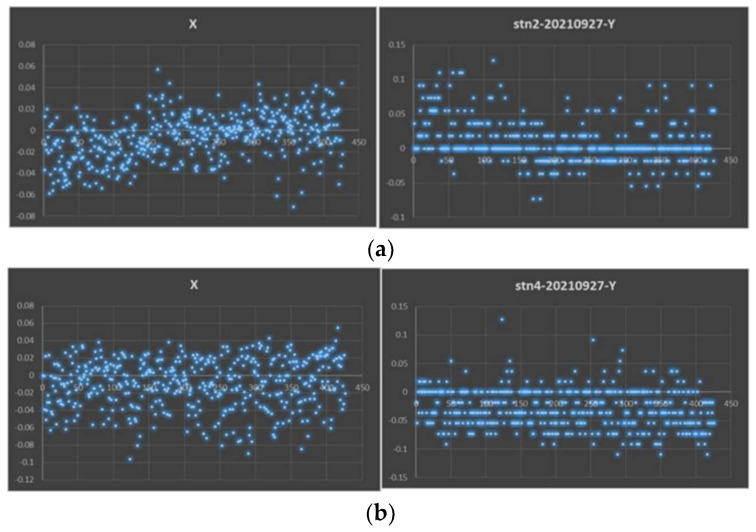
(**a**) Accuracy of AWC algorithm for STN2. (**b**) Accuracy of AWC algorithm for STN4.

**Figure 12 sensors-23-08502-f012:**
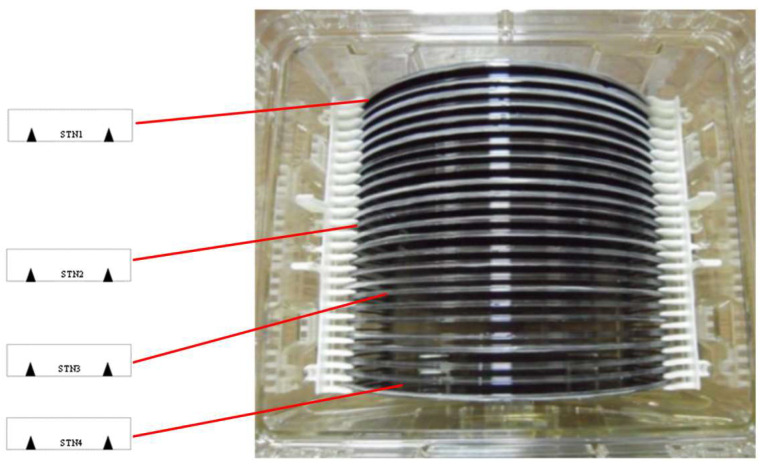
STN1~STN4 station.

**Figure 13 sensors-23-08502-f013:**
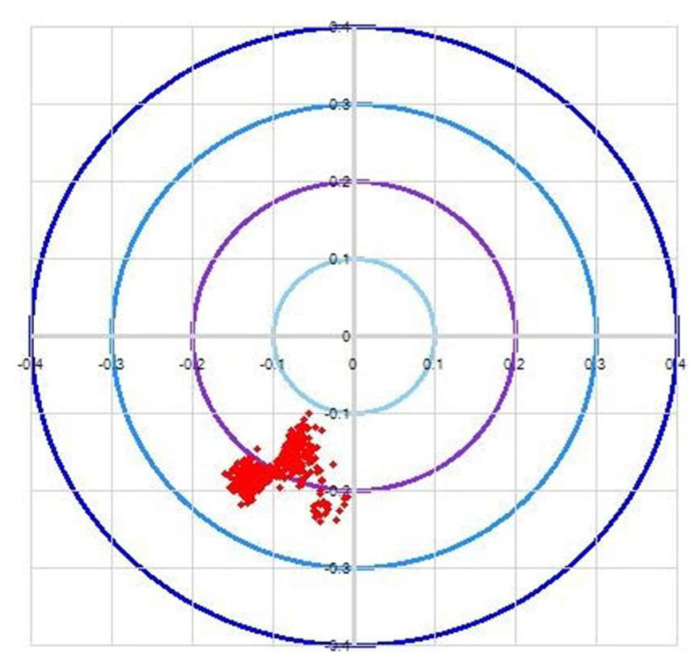
Accuracy of AWC algorithm for B-hand PICK wafers from 2 to 3 stations.

**Figure 14 sensors-23-08502-f014:**
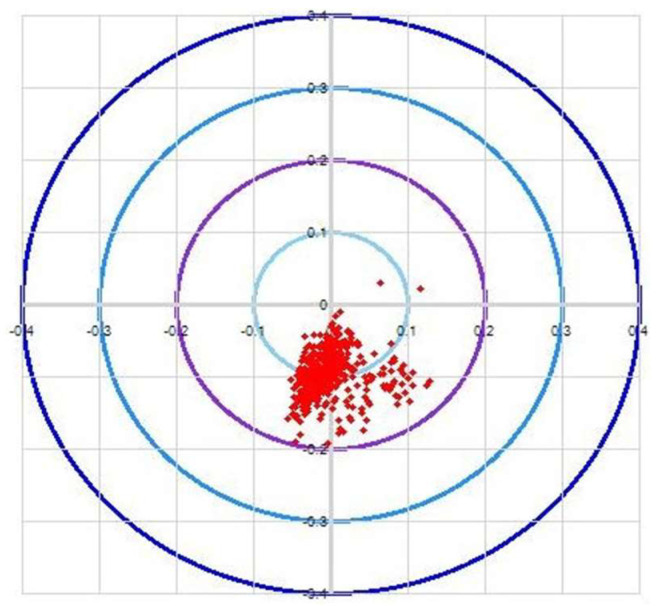
Accuracy of AWC algorithm for A-hand PICK wafers from 3 to 4 stations.

**Figure 15 sensors-23-08502-f015:**
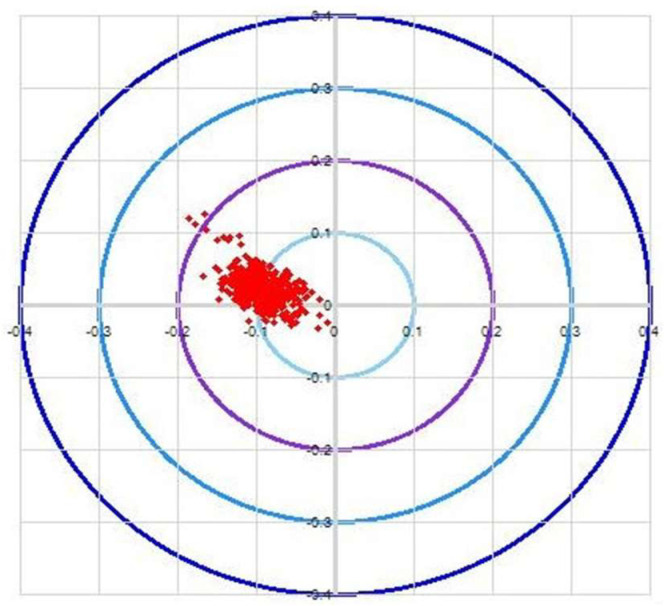
Accuracy of AWC algorithm for B-hand PICK wafers from 4 to 2 stations.

**Table 1 sensors-23-08502-t001:** Calibration of dual sensors during robot stretches.

Sensor Position	$P_{x}$ (mm)	$P_{y}$ (mm)
STN2 left	347.108654	436.242168
STN2 right	491.692988	285.199749
STN4 left	90.105124	520.304507
STN4 right	−119.096223	526.010610

**Table 2 sensors-23-08502-t002:** Calibration of dual sensors during robot retracts.

Sensor Position	$P_{x}$ (mm)	$P_{y}$ (mm)
STN2 left	347.191978	436.337168
STN2 right	491.819200	285.329218
STN4 left	90.123207	520.472600
STN4 right	−119.115173	526.250830

**Table 3 sensors-23-08502-t003:** Calculation of wafer deviation in finger coordinate system.

Position Correction	$D_{x}$ (mm)	$D_{y}$ (mm)
STN2 deviation	−0.810415	−0.960259
STN4 deviation	−0.862323	−0.976955

**Table 4 sensors-23-08502-t004:** Calculation of the expansion compensation value ΔR and the rotation compensation value ΔT.

Retraction Detection	ΔR (mm)	ΔT
STN2 compensation	0.959894	0.051550
STN4 compensation	0.976556	0.053054

## Data Availability

Not applicable.

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
