# Peer review of "Research on Motion Control and Wafer-Centering Algorithm of Wafer-Handling Robot in Semiconductor Manufacturing"

_sensors, 2023, doi:10.3390/s23208502_

Round 1

Reviewer 1 Report

The paper presents a wafer centering algorithm for a handling robot. The proposed algorithm and approach is documented analytically and validated in reality within a robotics workbench.

The paper is readable and comprehensive. The topic seems to be original and addresses a rare concept in the field.

The quality of the paper is about average.

However, some improvements are necessary.

Any related studies are not presented (e.g. in a comparative way).

The contributions are not sufficiently stated.

There are no sufficient details of the application software developed that implements the proposed algorithm.

A general view of the proposed algorithm (or procedures), e.g. as a flowchart perhaps would have been useful.

Figure 11(a) & (b) are not very clear (perhaps are too small).

I am not sure whether Figure 12 provides any useful information, but I do not insist, it may remain.

Some of the references are missing year of publication (perhaps some of them need some update).

English language is about fine.

Reviewer 2 Report

This study aims to develop the AWC (Active Wafer Centering) algorithm for the movement control and wafer calibration of the handling robot in semiconductor manufacturing. Although this article was employed with the benefit of enhancing precision for semiconductor industry, the academic contribution is not sufficient. It is recommended that the authors clarify their contribution for better comprehension. More specific major and minor comments are given below.

(1)  It is highly recommended to have native speakers and professional writing assistance for significant improvement. The reviewer had a really hard time to follow or understand the topic in the current work.

(2)  The literature review section in the introduction was lack of representative and recent studies in the research field. What are the current state-of-the-art and most influential studies? What are the current challenges in this field? The review of recent literatures is not sufficient.

(3)  The clarity of the optical sensor in this study should be enhanced, such as brand, type, specification even a customization sensor.

(4)  The resolution and readability of the figures need to be improved, such as figure 1, 3, 5b, 6 and 7. Further, the label of the figure also should be addressed, such as figure 9, 10, 11, 13, 14 and 15.

(5)  The purpose and meaning of figure 12 is unclear. It is a really hard time to understand the meaning in the figure 12.

(6)  How does the AWC (Active Wafer Centering) algorithm work with the control system? Or which layer of figure 2 can realize the implementation of AWC.

All in all, this a promising work. I hope that the author can help improve the quality of the paper.

It is highly recommended to have native speakers and professional writing assistance for significant improvement. The reviewer had a really hard time to follow or understand the topic in the current work.

Reviewer 3 Report

In this paper, the Authors are proposing the AWC (Active Wafer Centering) algorithm for the movement control and wafer calibration of the handling robot in semiconductor manufacturing.

The results show that the proposed method has a good response.

I have found the study really interesting and appropriate for this problem.

After carefully reading, I find that this paper is extremely interesting, and it’s ready for publishing.

Round 2

Reviewer 1 Report

No further comments.

English language is fine.

Reviewer 2 Report

The article has been significantly improved. It can be accepted in the revision.